# Real-World Evaluation of Universal Germline Screening for Cancer Treatment-Relevant Pharmacogenes

**DOI:** 10.3390/cancers13184524

**Published:** 2021-09-08

**Authors:** Megan L. Hutchcraft, Nan Lin, Shulin Zhang, Catherine Sears, Kyle Zacholski, Elizabeth A. Belcher, Eric B. Durbin, John L. Villano, Michael J. Cavnar, Susanne M. Arnold, Frederick R. Ueland, Jill M. Kolesar

**Affiliations:** 1Division of Gynecologic Oncology, Department of Obstetrics and Gynecology, University of Kentucky Markey Cancer Center, Lexington, KY 40536, USA; megan.hutchcraft@uky.edu (M.L.H.); fuela0@uky.edu (F.R.U.); 2Department of Pharmacy Practice and Science, University of Kentucky College of Pharmacy, Lexington, KY 40536, USA; nan.lin@uky.edu; 3Department of Pathology and Laboratory Medicine, University of Kentucky Chandler Medical Center, Lexington, KY 40536, USA; shulin.zhang@uky.edu (S.Z.); catherine.sears@uky.edu (C.S.); 4Department of Pharmacy, Virginia Commonwealth University Medical Center, Richmond, VA 23298, USA; kyle.zacholski@vcuhealth.org; 5Department of Clinical Research, University of Kentucky Markey Cancer Center, Lexington, KY 40536, USA; elizabeth.belcher@uky.edu; 6Division of Biomedical Informatics, Department of Internal Medicine, University of Kentucky College of Medicine, Lexington, KY 40536, USA; e.durbin1@uky.edu; 7Kentucky Cancer Registry, University of Kentucky Markey Cancer Center, Lexington, KY 40536, USA; 8Division of Medical Oncology, Department of Internal Medicine, University of Kentucky Markey Cancer Center, Lexington, KY 40536, USA; jlvillano@uky.edu (J.L.V.); susanne.arnold@uky.edu (S.M.A.); 9Division of Surgical Oncology, Department of Surgery, University of Kentucky Markey Cancer Center, Lexington, KY 40536, USA; michael.cavnar@uky.edu

**Keywords:** pharmacogenomics, cancer, real word, irinotecan, 5-fluorouracil, mercaptopurine

## Abstract

**Simple Summary:**

Germline pharmacogenomic variants impact the toxicity of many cancer treatment drugs. Though testing for pharmacogenomic variants prior to initiating systemic cancer treatment is not routine, the Clinical Pharmacogenetics Implementation Consortium recommends dosing modifications for six cancer treatment drugs based on variant genotypes: irinotecan and UGT1A1; 5-fluorouracil and capecitabine and DPYD; 6-mercaptopurine and thioguanine and TPMT; and tamoxifen and CYP2D6. The purpose of this study was to assess the frequency of cancer treatment-relevant germline pharmacogenomic variants in patients with cancer using residual germline whole-exome sequencing. We also evaluated the association of disease-relevant pharmacogenomic variants with treatment-associated toxicities. Approximately one-quarter of cancer patients carried a disease-relevant pharmacogenomic variant. Patients with toxicity-associated pharmacogenomic variant genotypes were more likely to experience drug-related toxicity than their wild-type counterparts. Universal pharmacogenomic screening is feasible using whole-exome sequencing originally obtained for quality control purposes and may be an effective germline pharmacogenomic screening strategy for patients who are candidates for irinotecan, 5-fluorouracil, capecitabine, or 6-mercaptopurine.

**Abstract:**

The purpose of this study was to determine the frequency of clinically actionable treatment-relevant germline pharmacogenomic variants in patients with cancer and assess the real-world clinical utility of universal screening using whole-exome sequencing in this population. Cancer patients underwent research-grade germline whole-exome sequencing as a component of sequencing for somatic variants. Analysis in a clinical bioinformatics pipeline identified clinically actionable pharmacogenomic variants. Clinical Pharmacogenetics Implementation Consortium guidelines defined clinical actionability. We assessed clinical utility by reviewing electronic health records to determine the frequency of patients receiving pharmacogenomically actionable anti-cancer agents and associated outcomes. This observational study evaluated 291 patients with cancer. More than 90% carried any clinically relevant pharmacogenetic variant. At least one disease-relevant variant impacting anti-cancer agents was identified in 26.5% (77/291). Nine patients with toxicity-associated pharmacogenomic variants were treated with a relevant medication: seven UGT1A1 intermediate metabolizers were treated with irinotecan, one intermediate DPYD metabolizer was treated with 5-fluorouracil, and one TPMT poor metabolizer was treated with mercaptopurine. These individuals were more likely to experience treatment-associated toxicities than their wild-type counterparts (*p* = 0.0567). One UGT1A1 heterozygote died after a single dose of irinotecan due to irinotecan-related adverse effects. Identifying germline pharmacogenomic variants was feasible using whole-exome sequencing. Actionable pharmacogenetic variants are common and relevant to patients undergoing cancer treatment. Universal pharmacogenomic screening can be performed using whole-exome sequencing data originally obtained for quality control purposes and could be considered for patients who are candidates for irinotecan, 5-fluorouracil, capecitabine, and mercaptopurine.

## 1. Introduction

Germline pharmacogenomic variants influence the metabolism of environmental toxins and affect patient response to medications. In 2009, the United States (US) Food and Drug Administration (FDA) added drug labeling for pharmacogenomic considerations [1], and the list now includes over 450 drugs [2]. The Clinical Pharmacogenetics Implementation Consortium (CPIC) helps clinicians and pharmacists navigate this complex genetic information and highlights the level of evidence supporting each pharmacogenomic variant’s importance [3,4]. CPIC provides evidence-based variant-specific prescribing guidance; currently, there are twenty pharmacogenes classified in level A status [3,4].

Oncology medications comprise 41.4% (189/457) of all medicines listed by the US FDA with pharmacogenomic considerations and 57.1% (108/189) of these medications are listed for reasons due to risk of adverse events, contraindications, boxed and other warnings, and medication precautions [2]. In addition, our institution previously reported that 65% of advanced cancer patients were taking a pharmacogenomically actionable medication [5]. 

Despite data suggesting benefit, pharmacogenomic screening is not routinely performed in clinical practice [6] due to logistical barriers, cost, and lack of clinical utility [7]. A possible solution to overcome logistical obstacles is the implementation of universal actionable pharmacogenomic variant screening [7]. Additionally, somatic mutation testing to guide treatment is now routine for many patients with cancer and paired somatic/germline sequencing is often performed to reliably distinguish between germline and somatic mutations. While initially used as a quality control measure to avoid reporting incidental germline findings [8], this germline testing may provide an opportunity to also assess pharmacogenomic variants using existing sequencing data [9,10]. Hertz and colleagues (2018) reported that the Michigan Oncology Sequencing program could determine germline TPMT, DPYD, and CYP2C19 genotype at no additional cost [11]. The purpose of this study is to determine the frequency of clinically actionable germline variants in patients with cancer and assess the real-world clinical utility of universal screening using germline whole-exome sequencing (WES) originally obtained for quality control purposes. 

## 2. Results

### 2.1. Study Population and Pharmacogenomic Landscape

Between October 2018 and January 2021, we prospectively enrolled 291 cancer patients, regardless of cancer type. Table 1 summarizes the demographic characteristics. The median age was 61 (inter-quartile range 52–68), and most individuals were non-Hispanic and White (277/291, 95.2%), with fourteen (14/291, 4.8%) non-Hispanic and Black individuals also participating. Approximately equal numbers of men and women participated.

Pharmacogenomic variants were common and are reported in Table 2. Overall, 90.4% (263/291) of cancer patients carried any pharmacogenomic variant. No CFTR, CYP2C19, CACNA1S, or HLA-B variants were detected; however, testing was limited for CFTR (*c.1652G>A only), CYP2C19 (*7 only), and HLA-B (*57:01, *15:02, and *58:01 only). A full listing of variants included in analysis is available in Appendix A. Although population allele frequencies vary by race and ethnicity, allele frequencies in our cohort were similar to expected American or European population frequencies [3] except for a higher than expected frequency of RYR1 variants, which was present in two patients and demonstrated an allele frequency of 0.0034.

### 2.2. Clinical Impact

Many (189/291, 64.9%) oncology patients harbored pharmacogenomic variants in genes that metabolize cancer therapeutic agents. Approximately one-quarter (26.5%, 77/291) carried at least one pharmacogenomic variant for at least one therapeutic option indicated in their disease type. Table 3 summarizes the association between pharmacogenes, their associated anti-cancer agents, National Comprehensive Cancer Network (NCCN) guideline-recommended anti-cancer drugs, and the number of patients at risk. Although most patients received systemic cancer treatments at our institution, several patients were treated elsewhere. Follow-up data were not available for six colorectal, one pancreatic, and one hepatobiliary cancer patient. 

Consistent with the expected American/European population frequency, UGT1A1 variants were common (157/291, 54%). UGT1A1 genotype status impacts irinotecan metabolism, often indicated for front-line treatment of gastrointestinal malignancies [14,15]. Over half (45/86, 52.3%) of the patients who could potentially receive irinotecan as a component of an NCCN “preferred” regimen harbored a UGT1A1 variant. The fluoropyrimidines capecitabine and fluorouracil are commonly included in cancer treatment regimens [16], and several patients (14/291, 4.8%) carried a DPYD variant, which affects the metabolism of these drugs. Similarly, TPMT is involved in the metabolism of 6-mercaptopurine (6-MP) [17], which is often administered in acute lymphoblastic leukemia (ALL) [18]; however, few ALL patients were enrolled in our study, and only one received 6-MP.

#### 2.2.1. Adverse Effects

Figure 1 illustrates the association between treatment-relevant genotype and treatment tolerance for patients prescribed CPIC level A drugs as a component of their cancer therapy. Treatment and tolerance details for each patient are available in Appendix B. Associated dosing of each treatment regimen is available in Appendix C. Of the six normal irinotecan metabolizers (UGT1A1 *1/*1), only one (1/6, 16.7%) experienced irinotecan-related diarrhea requiring a dose reduction. Five of seven (71.4%) UGT1A1 *28 heterozygotes (irinotecan intermediate metabolizers) experienced any irinotecan-related toxicity. Toxicities resulted in dose reductions (2/7, 14.3%), delays (1/7, 14.3%), and discontinuation (1/7, 14.3%); one patient received one cycle of irinotecan and presented with severe diarrhea, ultimately resulting in renal failure and death after a protracted hospital stay (1/7 14.3%). 

Similarly, the only DPYD intermediate metabolizer (*c.557A>G heterozygote) treated with a fluorouracil-containing regimen at our institution required treatment discontinuation after one cycle due to poor tolerance. Fluorouracil-related treatment toxicity was experienced by 40% (12/30) of normal DPYD metabolizers. Finally, the only patient in our cohort treated with 6-MP was not clinically tested prior to treatment and experienced profound cytopenia after the first dose. Genomic testing later revealed this patient was a poor TPMT metabolizer (TPMT *3B/*3C).

Though our statistical power is limited by a small sample size, patients with toxicity-associated variant alleles were more likely to experience toxicities resulting in treatment delays, dose reductions, or discontinuation of chemotherapy than those who did not carry a variant metabolism gene (7/9, 77.8% vs. 13/36, 36.1%; *p* = 0.0567) when treated with a pharmacogenomically-relevant drug.

#### 2.2.2. Efficacy

In our cohort, one intermediate tamoxifen metabolizer (CYP2D6*1/*6) was treated with tamoxifen. This patient experienced progressive disease with standard (20 mg daily) tamoxifen dosing and required alternative anti-estrogen treatment with an aromatase inhibitor and surgical castration. No additional patients were treated with tamoxifen for comparison purposes. 

## 3. Discussion

Our results suggest the feasibility and potential benefit of pharmacogenomic variant genotyping for cancer patients using WES obtained for quality control purposes in a real-world setting. Though our numbers are small, patients with treatment-relevant pharmacogenomic variants experienced increased rates of toxicity compared to normal metabolizers. FDA labeling does not require pharmacogenomic testing before initiating any CPIC level A cancer drugs [19]; however, our results suggest an increased risk of toxicity, including death, requiring clinical intervention for patients with treatment-relevant pharmacogenomic variants. 

Though variant-associated toxicity is established for TPMT [17], UGT1A1 [15], and DPYD [16] intermediate and poor metabolizers, testing is not routinely performed and clinical management guidelines for dosing are inconsistent. This is likely related to logistical barriers, lack of clinical utility, and cost associated with routine pharmacogenomic screening [7,11]. This research demonstrates germline WES data originally obtained for quality control purposes may be used to report pharmacogenomic variants. Germline sequencing data are increasingly available because of targeted treatment options and somatic mutation testing [10,11] and may represent a cost-effective strategy for both overcoming barriers to pharmacogenetic testing and integrating germline and somatic sequencing to expand precision cancer care.

Our results also demonstrate that pharmacogenomic variants are common; more than 90% of cancer patients carried at least one clinically actionable pharmacogenetic variant. Furthermore, our findings suggest potential clinical utility. In this unselected population, 15.8% (46/291) of individuals received a genotype-relevant anti-cancer drug and ten of these individuals carried a variant pharmacogene. Importantly, while our numbers are small, patients with treatment-relevant pharmacogenomic variants experienced increased rates of toxicity requiring clinical intervention and death compared to normal metabolizers. 

US FDA labeling does not require pharmacogenomic testing before initiating any CPIC level A cancer medications [19]. Regarding practice guidelines, ALL is the only disease in which the NCCN recommends, but does not require, testing for the presence of TPMT variants before 6-MP initiation [18]. The NCCN colon cancer guidelines advise caution and suggest alternative dosing strategies for UGT1A1 poor metabolizers (UGT1A*28 homozygous variants) scheduled to receive irinotecan; however, no testing guidelines have been established [19,20]. Conversely, these same guidelines note DPYD variants are inconsistently associated with fluoropyrimidine toxicity and do not currently recommend pre-treatment testing [20]. Despite the role of irinotecan, fluorouracil, and capecitabine in preferred NCCN treatment regimens for pancreatic [21] and gastroesophageal [22,23] cancers, these pharmacogenomic variants are not discussed in the guidelines.

The oncology community has hesitated to initiate lower fluoropyrimidine doses for patients with DPYD variants and lower irinotecan doses for UGT1A1 intermediate metabolizers because not all poor and intermediate metabolizers experience toxicity [15,16,24] and treating with a lower dose may decrease the drug’s therapeutic efficacy [24]. In fact, recent research demonstrated genotype-directed irinotecan dosing resulted in improved pathologic complete responses and decreased irinotecan-related toxicity for UGT1A1 intermediate metabolizers (*1/*28) undergoing neoadjuvant chemoradiation for locally advanced rectal cancer [25]. Similarly, DPYD genotype-directed fluoropyrimidine dosing demonstrated improved patient safety outcomes [26] without compromising efficacy in a diverse group of patients treated with capecitabine or fluorouracil containing regimens [27]. 

Avoiding treatment-related toxicity is critical; patients who experience treatment-related adverse events are more likely to discontinue treatment altogether [28], experience dose reductions and delays, and ultimately experience worse survival outcomes [29,30]. Additionally, preemptive pharmacogenomic screening for DPYD variants is a cost-effective approach for patients scheduled to receive fluoropyrimidine-based chemotherapy [31,32]; however, data remain mixed for assessing for TPMT variants in patients scheduled to receive 6-MP [33,34] and UGT1A1 variants in patients scheduled to receive irinotecan [35,36,37,38]. Therefore, we suggest an individualized genotype-directed dosing strategy for patients scheduled to receive a pharmacogenomically-relevant anti-cancer medication. 

Strengths of this study include a prospective population-level approach, novel use of quality control germline data for identifying pharmacogenes, and assessment of the real-world impact of pharmacogenes in patients with cancer. In addition, variant calling was performed in a Clinical Laboratory Improvement Amendments setting with annotation by a pathologist. 

A potential limitation is the use of research-grade sequencing, which typically has lower coverage than clinical sequencing; however, allele frequencies were consistent with expected population allele frequencies suggesting this limitation is of minimal practical consequence. As a single institutional study treating a largely rural and predominantly non-Hispanic White population, generalizability may be limited. Furthermore, pharmacogenomic allele frequencies vary by race and ethnicity and our screening strategy has not been tested in a diverse population. We also did not collect blood samples, precluding an analysis of drug exposure, metabolizer status, and toxicity. Though patients were prospectively enrolled in this study, clinical outcome data were collected retrospectively, limiting clinical actionability. Finally, as our institution is a tertiary care center, many patients enrolled in this study underwent surgical treatment at our institution but received chemotherapy elsewhere, limiting sample size and follow-up data. 

## 4. Materials and Methods 

### 4.1. Study Design and Data Sources

This observational cohort study compared research-grade germline WES results to clinical treatment data and patient outcomes. Patients treated at the University of Kentucky Markey Cancer Center enrolled in the Total Cancer Care (TCC)^®^ protocol. 

TCC is a multi-institutional prospective cohort study and comprises data collected from eighteen member institutions of the Oncology Research Information Exchange Network (ORIEN). This cancer precision medicine initiative was first developed by the Moffitt Cancer Center in Tampa, Florida, USA [39,40]. To date, over 315,000 participants have enrolled, undergone germline and tumor somatic sequencing, and agreed to lifetime follow-up [41].

Demographic and clinical data were obtained from the Kentucky Cancer Registry (KCR). Kentucky state statute (KRS 214.556) requires all cases of cancer diagnosed and/or treated in Kentucky to be reported to this registry. This population-based cancer registry reports demographic and clinical information, including the genetic data generated as a component of the TCC protocol. 

The Markey Cancer Center Cancer Research Informatics Shared Resource Facility (CRI SRF) served as the honest broker and distributed data stored in the KCR. A contractual agreement was previously established through M2GEN (Hudson, FL, USA) to permit data sharing between ORIEN/TCC and KCR. All data were fully anonymized prior to analysis. The final dataset for this study comprised the linked demographic, clinical, and genomic data from KCR and ORIEN/TCC. 

### 4.2. Study Population

Between October 2018 and January 2021, patients presenting to Markey Cancer Center for cancer treatment were invited to enroll in TCC. Treating physicians informed eligible patients about this study, and subjects were recruited during routine clinic visits. Eligibility criteria for TCC required patients to be at least 18 years of age and have a diagnosis of cancer. To be included in this analysis, each patient must have had germline WES results available. All subjects provided written informed consent prior to study enrollment. This study was conducted according to the guidelines of the Declaration of Helsinki, in accordance with the US Common Rule and was approved by the Institutional Review Board (IRB protocol code #44224, initial date of approval 28 June 2017 and modified to permit return of research results to enrolling physicians on 12 October 2018) at the University of Kentucky. Demographic variables included gender, race, and age at diagnosis. Clinical variables included primary cancer site, histology, and American Joint Committee on Cancer stage at diagnosis. Additional data elements, including treatment regimens, duration of therapy, and adverse effects, were retrospectively abstracted from the electronic health record under a separate IRB approval (IRB protocol code #51483, initial date of approval 3 June 2019). Data were collected and managed using Research Electronic Data Capture (REDCap) hosted at the University of Kentucky [42,43]. 

### 4.3. Sequencing Methods

Buccal swabs were used to obtain samples for germline testing. Germline WES was performed as a quality control measure as a component of the ORIEN/TCC protocol. Preparation of M2GEN WES libraries involved hybrid capture using an enhanced Integrated DNA Technology (Coralville, IA, USA) WES kit (38.7 megabases) with additional custom-designed probes for double coverage of 440 cancer genes. Library hybridization is performed at either single or 8-plex and sequenced on an Illumina (San Diego, CA, USA) NovaSeq 6000 instrument generating 100 base pair paired reads. WES is performed on tumor/normal matched samples with the normal covered at 100X and the tumor covered at 300X (additional 440 cancer genes covered at 600X) depth. We performed both tumor/normal concordance and gender identity quality control checks. The minimum threshold for hybrid selection was >80% of bases with >20X fold coverage; M2GEN WES libraries typically met or exceeded 90% of bases with >50X fold coverage for tumor and 90% of bases with >30X fold coverage for normal samples. 

### 4.4. Pharmacogenomic Variant Interpretation and Reporting of Results 

An in-house bioinformatics pipeline was developed for analyzing and annotating germline variants in fifteen of the twenty genes with CPIC level A drug recommendations. NUDT15 and HLA-A variants were not included because probe coverage was suboptimal. IFNL3 and IFNL4 variants were not included because of the uncommon clinical usage of peginterferon alfa-2a and 2b. Haplotypes for each pharmacogene included in the testing process are reported in Appendix A. 

Briefly, FASTQ files generated from an Illumina sequencer were aligned to the reference sequence of human genome (GRCh37) using Burrows–Wheeler Aligner (BWA 0.7.8) [44]. Aligned reads were converted to Binary Alignment Map format using Sequence Alignment/Map tools software (V1.8) [45]. Variant calling was carried out using Genome Analysis Toolkit (V4.0.12.0) [46] and VarScan (v2.3.9) [47]. Variants were annotated using Ensembl Variant Effect Predictor (VEP_89) [48] and public databases, including ClinVar [49], 1000 Genomes [50], and the Genome Aggregation Database (gnomAD) [12]. 

Results of pharmacogenomic testing, including pharmacogene, variant allele, and zygosity were reported to the Markey Cancer Center precision medicine team. This team reviewed results and reported findings in a letter to the enrolling oncologist, which was uploaded into the electronic medical record. 

### 4.5. Assessment of Potential and Actual Risk for Pharmacogenomic Adverse Outcomes

CPIC defines level A evidence for drug–gene pairs where the preponderance of the evidence is moderate or strong in favor of making drug prescribing changes based on a pharmacogenomic marker [3]. In this study, we defined anti-cancer agents as medications intended to treat cancer. Based on these criteria, there are currently six anti-cancer agents with a CPIC level A recommendation: irinotecan and UGT1A1; 5-fluorouracil and capecitabine and DPYD; 6-MP and thioguanine and TPMT; and tamoxifen and CYP2D6. To determine the number of patients potentially at risk for anti-cancer agent and pharmacogene-associated adverse effects, we first reviewed NCCN guidelines for individual tumor types included in our population for “preferred” and “other” regimens (Appendix D) that included CPIC level A anti-cancer agents. Drugs included in “regimens useful in certain instances” for specified malignancy were included in NCCN “other” regimens. Patients with disease types with a “preferred” or “other” regimen specified in NCCN guidelines [51] were considered potentially impacted by a pharmacogenomic variant.

The actual impact of pharmacogenomic variants on cancer treatment was determined by assessing dosing changes, discontinuation, hospitalization, or treatment-related death resulting from known adverse effects associated with variant genotypes. These included cytopenia and diarrhea for irinotecan; cytopenia, diarrhea, hepatotoxicity, mucositis/stomatitis, and cardiotoxicity for fluoropyrimidines; and cytopenia for 6-MP. Specifically, allergic reactions, fatigue, or patient requests resulting in drug discontinuation or dose reductions were not considered dose- or genotype-related. Patients were evaluated for toxicity by each CPIC level A drug and by individual genotype and, as a result, may have been included in more than one toxicity analysis. 

### 4.6. Statistical Analysis 

We performed a descriptive analysis of clinical variable and genotype frequencies. Expected monoallelic population pharmacogene frequencies were obtained from CPIC [3] when available. When unavailable, the population expected frequencies were obtained from the Exome Aggregation Consortium (ExAC) database [13] or gnomAD [12]. Expected American population frequencies were prioritized, followed by expected European population frequencies as the Kentucky population is 84% non-Hispanic White [52] and allele frequencies vary by race and ethnicity. Fisher’s exact test was used to compare genotype and frequency of treatment-related toxicity.

## 5. Conclusions

Pharmacogenomically actionable variants are common in patients with cancer and were identified using a whole-exome sequencing approach. Individuals receiving relevant anti-cancer drugs experienced more toxicities than wild-type individuals. Therefore, universal pharmacogenomic screening using this approach could be considered for patients who are candidates for irinotecan, 5-fluorouracil, capecitabine, and 6-MP. 

## Figures and Tables

**Figure 1 cancers-13-04524-f001:**
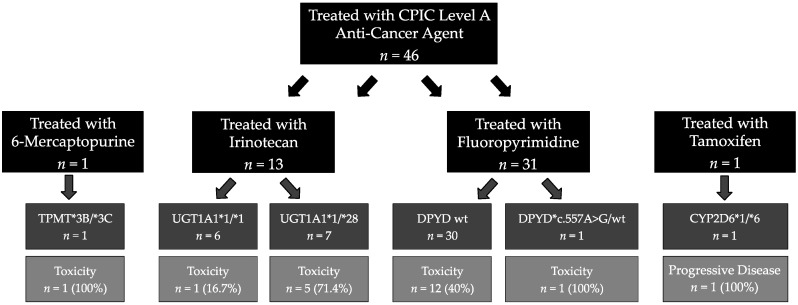
Patient tolerance to genomically-relevant anti-cancer agents. Toxicities resulting in dosing modifications, cycle delays, or discontinuation are reported. 6-mercaptopurine toxicity included cytopenia; irinotecan toxicities included cytopenia and diarrhea; fluoropyrimidine toxicities included cytopenia, diarrhea, hepatotoxicity, mucositis/stomatitis, and cardiotoxicity. Disease response is reported for the one patient who received tamoxifen as CYP2D6 is an efficacy-related pharmacogene. Abbreviations: CPIC: Clinical Pharmacogenetics Implementation Consortium; wt: wild type.

**Table 1 cancers-13-04524-t001:** Patient demographics and disease characteristics.

Characteristic	Patients (*n*)	%
Total	291	
Age (median)	61	IQR: 52–68
Race		
Non-Hispanic Black	14	4.8%
Non-Hispanic White	277	95.2%
Gender		
Female	147	50.5%
Male	139	47.8%
Non-Binary ^1^	1	0.3%
Primary Cancer Site		
Colon/rectal	55	18.9%
Gynecologic	49	16.8%
Head and neck	36	12.4%
Brain	25	8.6%
Pancreatic	16	5.5%
Leukemia/lymphoma	15	5.2%
Gastric/gastroesophageal	15	5.2%
Kidney and bladder	14	4.8%
Lung	14	4.8%
Small bowel ^2^	12	4.1%
Skin	12	4.1%
Breast	7	2.4%
Other ^3^	21	7.2%
Cancer Stage		
I	31	10.7%
II	42	14.4%
III	93	32.0%
IV	78	26.8%
N/A ^4^	47	16.2%

Abbreviations: IQR: inter-quartile range; N/A: not available or not applicable. ^1^ One patient identified as a transgender and was assigned male at birth. ^2^ Small bowel histologies included adenocarcinoma, carcinoid/neuroendocrine tumor, and sarcoma. ^3^ Other primary sites included anus, peripheral nervous system, prostate, soft tissue, and thyroid gland. ^4^ N/A includes cancers that are not staged or whose stage was unavailable.

**Table 2 cancers-13-04524-t002:** Pharmacogenomic variants and patient frequencies. The number of patients and percentage refers to heterozygotes unless otherwise specified.

Pharmacogenomic Variant	Patients with Variant Allele *n* (%)	Variant Allele Frequency	Expected Variant Allele Frequency ^1^
**Any**	263 (90.4%)		
**UGT1A1**			
*28		0.3093	0.3165
Homozygote	23 (7.9%)		
Heterozygote	134 (46.0%)		
Total UGT1A1	157 (54.0%)	0.3093	0.3165
**DPYD**			
*c.2846A>T	1 (0.3%)	0.0017	0.0037
*2A	4 (1.4%)	0.0069	0.0079
*HapB3	7 (2.4%)	0.0120	0.0237
*c.557A>G	1 (0.3%)	0.0017	0.0001
*7	1 (0.3%)	0.0017	0.0002
Total DPYD	14 (4.8%)	0.0240	0.0353
**TPMT**			
*3A		0.0601	0.0343
Homozygote	2 (0.7%)		
Heterozygote	31 (10.7%)		
*3B	1 (0.3%)	0.0017	0.0027
*3C		0.0123	0.0047
Homozygote	1 (0.3%)		
Heterozygote	3 (1.0%)		
*2	2 (0.7%)	0.0034	0.0021
Total TPMT	40 (14.0%)	0.0775	0.0438
**CYP2D6**			
*6	5 (1.7%)	0.0086	0.0025
**CYP2C9**			
*3	37 (12.7%)	0.0636	0.0301
*11	2 (0.7%)	0.0034	0.0028
Total CYP2C9	39 (13.4%)	0.0670	0.0329
**CYP3A5**			
*6	3 (1.0%)	0.0052	0.0015
*7	2 (0.7%)	0.0034	0.0000
Total CYP3A5	5 (1.7%)	0.0086	0.0015
**G6PD**			
A-202A_376G-III	1 (0.3%)	0.0017	0.0000–0.0340 ^2^
**CYP4F2**			
*3		0.2629	0.4108
Homozygote	21 (7.2%)		
Heterozygote	111 (38.1%)		
Total CYP4F2	132 (45.0%)		
**SLCO1B1**			
*15 or *17 ^3^		0.1186	0.1214 (*15); 0.0519 (*17)
Homozygote	5 (1.7%)		
Heterozygote	59 (20.3%)		
*5		0.0241	0.0224
Homozygote	2 (0.7%)		
Heterozygote	10 (3.4%)		
Total SLCO1B1	12 (4.1%)	0.1427	0.1957
**VKORC1**			
*1173C>T		0.1409	0.4643
Homozygote	11 (3.8%)		
Heterozygote	60 (20.6%)		
Total VKORC1	71 (24.4%)		
**RYR1**			
c.7042_7044delGAG	1 (0.3%)	0.0017	0.0000 ^4^
c.14818G>A	1 (0.3%)	0.0017	0.0000 ^5^
Total RYR1	2 (0.6%)	0.0034	0.0000 ^4,5^

^1^ Expected American population variant allele frequencies were obtained from CPIC database [3] unless otherwise specified. When expected American population variant frequency was not available, expected European population frequencies were reported. ^2^ Caucasian prevalence of this G6PD variant is 0.0000; however, prevalence of any G6PD variant in the Americas is 0.0340. ^3^ Testing bait was unable to differentiate SLCO1B1 *15 and *17 variants, so these frequencies were combined. ^4^ Population allele frequency for this variant was obtained from the Genome Aggregation Database (gnomAD) [12]. ^5^ Population allele frequency for this variant was obtained from the Exome Aggregation Consortium (ExAC) database [13].

**Table 3 cancers-13-04524-t003:** Pharmacogenes and their clinical relevance to cancer patients. Clinical Pharmacogenetics Implementation Consortium (CPIC) level A pharmacogenes are listed with their associated anti-cancer drugs and associated malignancies as per National Comprehensive Cancer Network (NCCN) guidelines. Patients at potential risk indicates patients with any disease-relevant pharmacogenomic variants for each specified malignancy regardless of treatment received. Numerator indicates patients with variant alleles and denominator includes all patients with specified malignancy. In other words, of the 55 patients with colon/rectal cancer, 33 had UGT1A1 variants. Patients at actual risk indicates patients with variant alleles for each associated malignancy who were treated with associated anti-cancer drugs at our institution. Numerator indicates patients treated with specified drug and denominator includes all patients with specified malignancy with associated variant allele. In other words, of the 55 patients with colon/rectal cancer, 27 patients had a UGT1A1 variant and were treated at our institution; five of those patients received irinotecan.

Pharmacogene	Anti-Cancer Drug	Malignancy	Patients at Potential Risk (a ^2^/A ^3^)	Patients at Actual Risk (b ^4^/B ^5^)
**Toxicity-Associated Pharmacogenes**				
UGT1A1	Irinotecan	Colon/rectal ^P^	33/55	5/27 ^1^
		Pancreas ^P^	7/16	2/6 ^1^
		Gastric/gastroesophageal ^P^	4/15	0/4
		Cervix ^O^	1/1	0/1
		Ovary ^O^	16/25	0/16
		Hepatobiliary ^O^	5/7	0/4 ^1^
		Carcinoid/neuroendocrine ^O^	6/10	0/6
		Small bowel adenocarcinoma ^O^	0/2	N/A ^6^
DPYD	Capecitabine	Colon/rectal ^P^	3/55	0/2 ^1^
		Pancreas ^P^	1/16	0/1
		Breast ^P^	0/7	N/A
		Ovary ^O^	2/25	0/2
		Cervix ^O^	0/1	N/A
		Anus ^O^	0/1	N/A
		Bladder ^O^	0/3	N/A
DPYD	Fluorouracil	Colon/rectal ^P^	3/55	0/2 ^1^
		Pancreas ^P^	1/16	1/1
		Gastric/gastroesophageal ^P^	1/15	1/1
		Head and neck ^P^	0/36	N/A
		Anus ^O^	0/1	N/A
		Vulva ^O^	0/1	N/A
		Basal cell skin ^O^	0/1	N/A
		Squamous cell skin ^O^	0/4	N/A
		Bladder ^O^	0/3	N/A
		Thyroid ^O^	0/4	N/A
		Small bowel adenocarcinoma ^O^	0/2	N/A
TPMT	Mercaptopurine	Acute lymphoblastic leukemia ^P^	1/3	1/1
**Efficacy-Associated Pharmacogenes**				
CYP2D6	Tamoxifen	Breast ^P^	1/7	1/1
		Ovary ^P^	1/25	0/1
		Uterus ^P^	0/22	N/A

Abbreviations: N/A: not available. ^P^ Drug is a component of an NCCN “preferred regimen” for specified malignancy. ^O^ Drug is a component of an NCCN “other regimen” and/or “regimen useful in certain instances” for specified malignancy. ^1^ Chemotherapy regimen was unknown for several patients who received treatment at outside institutions. ^2^ Number of patients with any variant allele. ^3^ Number of patients with this disease type. ^4^ Number of patients treated with drug; individuals treated outside of our institution are excluded. ^5^ Number of patients with any variant allele specific to drug; individuals treated outside of our institution are excluded. ^6^ If no patients with variant alleles were treated at our institution, ‘N/A’ is indicated.

## Data Availability

The authors received no special privileges in accessing the data. Raw data cannot be shared because they are both potentially identifying and contain sensitive patient data, including geographic location and specific dates of diagnosis, testing, and receipt of a drug. Additionally, there are contractual agreements between the University of Kentucky and the Kentucky Cancer Registry that preclude data sharing. Any requests for data must be submitted to: Jacyln K. McDowell, Epidemiologist, Kentucky Cancer Registry 2365 Harrodsburg Rd, Suite A230 Lexington, KY 40504; phone: (859)-218-2228; email: Jacyln.McDowell@uky.edu.

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
