# Peer review of "Real-World Evaluation of Universal Germline Screening for Cancer Treatment-Relevant Pharmacogenes"

_cancers, 2021, doi:10.3390/cancers13184524_

Round 1

Reviewer 1 Report

The submitted manuscript represents several important findings, was well designed, and is generally well written. There are a few changes which the author could explore which would improve the manuscript. These suggested changes are noted below.

Major comments:

  • This manuscript provides a thorough analysis of germline pharmacogenomics of anti-cancer agents. As a large portion of the applications of pharmacogenomics for cancer patients may include supportive care medications, I would recommend making it more clear in the title and abstract that this data applies specifically to anti-cancer medications.
  • Given the correlation between drug exposure and risk of toxicity, could the authors explore time/exposure to event analysis by metabolizer status?
  • Given the findings of this study, along with previously published data, I would suggest revising the focus of the discussion section. The data presented here does not provide significant evidence that pharmacogenomic testing should or should not be conducted in certain sub-populations of patients with cancer. Rather, it demonstrates the feasibility of incorporating pharmacogenomic analysis of NGS data (similar to PMID: 32832831). I believe that the importance of your data is in how it was obtained rather than the relatively small clinical signals identified. The clinical signals mentioned are important and show the potential of this strategies, but likely should not be the primary focus of the paper.

Minor comments:

  • Introduction
    • I would be sure to include and cite previous efforts to incorporate germline pharmacogenomic sequencing analysis from cancer sequencing data.
  • Methods
    • Please provide further insight if there are notable pharmacogenomic variants that would not be picked up on WES.
    • This study is noted to be prospective. It is unclear if the study was conducted prospectively or was a retrospective analysis of data which would have been collected prospectively. Please provide further clarification.
    • Given potential issues with quality DNA collection and integrity from buccal swabs in cancer patients, please provide a summary of quality control metrics for the data used in this study.
    • Given that you are not directly testing for pharmacogenomic variants but are repurposing the germline “control” meant for somatic testing, would you please expand on the bioinformatic steps necessary to get this data and incorporate into clinical care?
  • Results
    • It would be helpful to include a healthcare utilization analysis given these patients were seen in a single system.
    • Please provide a brief outline in Results 2.1 to explain the number of patients in this cohort treated internally and outside of institution.
    • Section 2.2.1., remove “numerically” from the sentence “Patients with treatment-relevant pharmacogenomic variants were numerically more likely to experience treatment toxicities that resulted in delays, dose reductions, or discontinuation of chemotherapy than those who did not carry a variant metabolism gene. If the population size is not powered to fide a statistical difference between groups with a p<0.05, it would be preferable to note.
    • Table 3 is difficult to read. As an example, it seems that two patients in the cohort received tamoxifen (one breast one ovary) with one of them having a variant allele. Later in the test the variant allele patient was discussed but the text mentions “No additional patients were treated with tamoxifen for comparison purposes”. Further refinement of the table is needed to ensure that the data is communicated clearly.
    • There is an extra “.” At the end of the legend for table 3.
    • When comparisons are made throughout the results section, please ensure that the criteria for inclusion in the comparison (disease, drug, dose, route, ect) is clear. This is difficult given the design of the study but necessary to understand what you are comparing.
    • Table 4 is difficult to digest. Would recommend making this table supplementary and including a visual summary of the data.
  • Discussion
    • Recommend defining results as demonstrating benefit of pharmacogenomics to potential benefit as proving benefit was not possible with the design of this study.
  • Conclusions
    • I think you should consider your results as supporting use of existing germline WES for pharmacogenomics in cancer patients instead of supporting pharmacogenomic testing in general. Your clinical data is not as strong as previous reports supporting pharmacogenomic testing. The method of obtaining your data is novel and should be of greater focus.

Author Response

The submitted manuscript represents several important findings, was well designed, and is generally well written. There are a few changes which the author could explore which would improve the manuscript. These suggested changes are noted below.

Author response: We appreciate the reviewer’s comments and the time to provide us with valuable feedback for our manuscript.

Major comments:

This manuscript provides a thorough analysis of germline pharmacogenomics of anti-cancer agents. As a large portion of the applications of pharmacogenomics for cancer patients may include supportive care medications, I would recommend making it more clear in the title and abstract that this data applies specifically to anti-cancer medications.

Author response: The title was revised as follows (new sections underlined): Real-World Evaluation of Universal Germline Screening for Cancer Treatment-Relevant Pharmacogenes. Additionaly, our abstract has been updated to better reflect the scope of our paper.

Given the correlation between drug exposure and risk of toxicity, could the authors explore time/exposure to event analysis by metabolizer status?

Author Response: Thank you for bringing up this important point. We agree it would be a relevant to perform this analysis; unfortunately, we did not collect blood samples from patients when they were receiving relevant medications and are unable to perform this analysis. We have added this as an additional limitation.

Given the findings of this study, along with previously published data, I would suggest revising the focus of the discussion section. The data presented here does not provide significant evidence that pharmacogenomic testing should or should not be conducted in certain sub-populations of patients with cancer. Rather, it demonstrates the feasibility of incorporating pharmacogenomic analysis of NGS data (similar to PMID: 32832831). I believe that the importance of your data is in how it was obtained rather than the relatively small clinical signals identified. The clinical signals mentioned are important and show the potential of this strategies, but likely should not be the primary focus of the paper.

Author response: Thank you for this feedback. We have updated the Abstract, Introduction, and Discussion to reflect this recommendation.

Minor comments:

Introduction

I would be sure to include and cite previous efforts to incorporate germline pharmacogenomic sequencing analysis from cancer sequencing data.

Author response: Thank you for this feedback. We have updated the Introduction to acknowledge previous efforts incorporating germline pharmacogenomic sequencing from cancer sequencing data.

Methods

Please provide further insight if there are notable pharmacogenomic variants that would not be picked up on WES.

Author response: Section 4.4 describes CPIC Level A pharmacogenes that were not included due to suboptimal coverage or clinical relevance. Though other publications suggest a “minimum” number of alleles for a pharmacogenetic testing panel (van der Wouden et al. 2019, PMID: 31038729 and Pratt et al. 2020, PMID: 32380173), currently, there is no universally accepted minimum number of genes and alleles for clinical pharmacogenetic testing. For this study, we reported the detectable clinically relevant alleles in Appendix A.

This study is noted to be prospective. It is unclear if the study was conducted prospectively or was a retrospective analysis of data which would have been collected prospectively. Please provide further clarification.

Author response: Patients were prospectively enrolled onto the Total Cancer Care protocol; however, data collection was retrospective after enrollment. This has been clarified in the Methods and Discussion.

Given potential issues with quality DNA collection and integrity from buccal swabs in cancer patients, please provide a summary of quality control metrics for the data used in this study.

Author response: Regarding quality control, our minimum threshold for variant calling is 20x coverage after passing the general sequencing quality control mentioned in Section 4.3 “Sequencing Methods.” 

Given that you are not directly testing for pharmacogenomic variants but are repurposing the germline “control” meant for somatic testing, would you please expand on the bioinformatic steps necessary to get this data and incorporate into clinical care?

Author Response: Additional details regarding the bioinformatic pipeline and incorporation into clinical care is detailed in Section 4.4.

Results

It would be helpful to include a healthcare utilization analysis given these patients were seen in a single system.

Author Response: Thank you for this comment. We agree it is an important future direction.  

Please provide a brief outline in Results 2.1 to explain the number of patients in this cohort treated internally and outside of institution.

Author Response: The number of patients who were not treated at our institution has been clarified in the last sentence of the first paragraph of Section 2.2. Clinical Impact.

Section 2.2.1., remove “numerically” from the sentence “Patients with treatment-relevant pharmacogenomic variants were numerically more likely to experience treatment toxicities that resulted in delays, dose reductions, or discontinuation of chemotherapy than those who did not carry a variant metabolism gene. If the population size is not powered to fide a statistical difference between groups with a p<0.05, it would be preferable to note.

Author Response: “Numerically” has been removed from this sentence. A comment regarding sample size and power has been placed at the beginning of this sentence.

Table 3 is difficult to read. As an example, it seems that two patients in the cohort received tamoxifen (one breast one ovary) with one of them having a variant allele. Later in the test the variant allele patient was discussed but the text mentions “No additional patients were treated with tamoxifen for comparison purposes”. Further refinement of the table is needed to ensure that the data is communicated clearly.

Author Response: We have made changes to the table caption for clarification purposes. The second from the right column indicates patients with each malignancy who have variant alleles and does not take into account actual treatments received (these patients are considered to be at potential risk) whereas the patients in the far right column are patients with each malignancy with variant alleles who were actually treated with the drug (these patients are considered to be at actual risk).

There is an extra “.” At the end of the legend for table 3.

Author Response: The extra “.” has been removed.

When comparisons are made throughout the results section, please ensure that the criteria for inclusion in the comparison (disease, drug, dose, route, ect) is clear. This is difficult given the design of the study but necessary to understand what you are comparing.

Author Response: This has been clarified. Changes have been made in Section 2.2.1 Adverse Effects.

Table 4 is difficult to digest. Would recommend making this table supplementary and including a visual summary of the data.

Author Response: Thank you for your feedback. This table has been moved to the appendix and a visual summary of the data is now present as Figure 1.

Discussion

Recommend defining results as demonstrating benefit of pharmacogenomics to potential benefit as proving benefit was not possible with the design of this study.

Author Response: This has been redefined in the first sentence of the Discussion, Section 3.

Conclusions

I think you should consider your results as supporting use of existing germline WES for pharmacogenomics in cancer patients instead of supporting pharmacogenomic testing in general. Your clinical data is not as strong as previous reports supporting pharmacogenomic testing. The method of obtaining your data is novel and should be of greater focus.

Author Response: Thank you for this comment. We have updated the Discussion to reflect your recommendations.

Reviewer 2 Report

The original is very interesting, the method is appropriate, the results are adequately expressed, including the tables. The conclusions, and the final proposal are relevant.

There are only minor issues that should be considered by the authors

  1. The method appears as point 4 in the construction of the article and perhaps it should be included as point 2 after the introduction
  2. The 95,2% of patients included are white, indeed the conclusions of the study cannot be applied to all populations. The authors state in the discussion that this issue is a limitation, but they should delve more deeply into the hypothetical differences according to race found in the references.
  3. The economic impact of applying the Universal pharmacogenomic screening for patients who are candidates for irinotecan, 5-fluorouracil, capecitabine, and mercaptopurine should be analyzed in the discussion.

Author Response

The original is very interesting, the method is appropriate, the results are adequately expressed, including the tables. The conclusions, and the final proposal are relevant.

There are only minor issues that should be considered by the authors

Author Response: Thank you for your comments. We appreciate the time you have taken to improve our manuscript.

The method appears as point 4 in the construction of the article and perhaps it should be included as point 2 after the introduction

Author Response: We followed the pre-specified formatting of the Cancers journal template, which includes the following order: 1 – Introduction, 2 – Results, 3 – Discussion, 4 – Materials and Methods. Please notify us if a different formatting should be used and we are happy to oblige. 

The 95,2% of patients included are white, indeed the conclusions of the study cannot be applied to all populations. The authors state in the discussion that this issue is a limitation, but they should delve more deeply into the hypothetical differences according to race found in the references.

Author Response: Thank you for your comment. We have incorporated further acknowledgement of the variability of pharmacogenomic allele frequencies by race / ethnicity in the Results Section 2.1 and in the Discussion Section 3.

The economic impact of applying the Universal pharmacogenomic screening for patients who are candidates for irinotecan, 5-fluorouracil, capecitabine, and mercaptopurine should be analyzed in the discussion.

Author Response: The economic impact of pharmacogenomic screening is reviewed in the Discussion. We mention that pre-emptive pharmacogenomic screening for DPYD variants has been demonstrated to be cost-effective for patients scheduled to receive fluoropyrimidine-based chemotherapy. We also discuss that data regarding the economic impact of screening for TPMT and UGT1A1 variants are mixed and discuss that future evaluation is warranted.
